# Determination of the Optimal Landmark for Tube Thoracostomy in Trauma Patients: A Retrospective Study

**DOI:** 10.3390/jcm14217571

**Published:** 2025-10-25

**Authors:** Mina Lee, Jaeik Jang, Jae-Hyug Woo, Hyuk Jun Yang, Woo Sung Choi, Jae Ho Jang, Sung Youl Hyun

**Affiliations:** 1Department of Emergency and Critical Care Medicine, Gachon University Gil Medical Center, Incheon 21565, Republic of Korea; 2Department of Traumatology, Gachon University Gil Medical Center, Incheon 21565, Republic of Korea; 3Gachon University College of Medicine, Incheon 21565, Republic of Korea

**Keywords:** chest tubes, thoracostomy, sternum, diaphragm, pneumothorax, hemothorax, methods, thoracic injuries, tomography, X-ray computed

## Abstract

**Background/Objectives**: Accurate and prompt tube thoracostomy (TT) placement within the safety zone while avoiding diaphragmatic injury remains challenging, particularly in trauma patients with distorted thoracic anatomy. This study evaluated the accuracy and safety of landmark-based TT techniques, including a novel mid-sternum method. **Methods**: In this retrospective study, chest computed tomography scans of 245 adult trauma patients who presented to a Level I trauma center in Korea between February and June 2022 were analyzed. TT insertion routes using the mid-sternum, nipple, and mid-arm point methods were compared against the conventional fifth intercostal space (ICS) method. **Results**: Of the 245 enrolled patients, the median age was 55.0 years (interquartile range, 42.0–64.0), and 186 (75.9%) were male. On the right side, routes avoiding the diaphragm were observed in 82.0% (fifth ICS), 92.7% (mid-sternum), 55.5% (nipple), and 90.2% (mid-arm point) of patients. The mid-sternum method showed a significantly higher avoidance rate than the fifth ICS method (*p* < 0.001), with 91.1% sensitivity and 77.4% specificity for identifying TT routes within the safety zone. On the left side, routes avoiding the diaphragm were observed in 97.6% (fifth ICS), 98.8% (mid-sternum), 86.9% (nipple), and 95.1% (mid-arm point) of patients, with no significant difference between the fifth ICS and mid-sternum methods (*p* = 0.375). The mid-sternum method showed 90.4% sensitivity and 85.2% specificity for routes within the safety zone. **Conclusions**: The mid-sternum method demonstrated high anatomical safety and performance comparable to or superior to the conventional fifth ICS method, particularly in minimizing the risk of diaphragmatic injury. It may offer a practical and safe alternative for TT placement in trauma care.

## 1. Introduction

Tension pneumothorax and traumatic hemothorax are life-threatening emergencies that require prompt chest tube insertion. Chest tube thoracostomy is generally recommended within the “safe triangle”, which is defined by the anterior border of the pectoralis major, the lateral border of the latissimus dorsi, the inferior border of the fifth intercostal space (ICS), and the base of the axilla [1]. To comply with this safety zone, current major guidelines recommend chest tube insertion at the intersection of the fifth ICS and the mid-axillary line (MAL), just anterior to it [2].

Despite these clear guidelines, an observational simulation-based study reported that only 51% of intercostal drain insertion sites identified by participants were located within the recommended “triangle of safety”, underscoring the ongoing risk of malposition [3]. In patients with severe chest trauma, anatomical distortions such as rib fractures may obscure conventional landmarks and increase the risk of improper tube placement [4]. Furthermore, in emergency situations—particularly in obese patients—the presence of thick subcutaneous fat layers may make it challenging to identify the appropriate insertion site quickly [5]. Life-threatening emergencies, such as tension pneumothorax, demand rapid and accurate chest tube placement, which can be challenging when surface landmarks are not readily discernible.

When anatomical landmarks are not easily identifiable, needle decompression is often attempted as an initial step [2]. However, this procedure has several limitations, including restricted air release, catheter kinking or obstruction, difficulty identifying anatomical structures, and an increased risk of infection. According to Wernick et al., the success rate of needle decompression ranges from 68% to 75% [6], while Ball et al. reported failure rates of up to 65% when short catheters were used [7]. Ultrasound-guided chest tube insertion has been studied as an alternative to improve anatomical localization and insertion accuracy [8,9]. Nevertheless, ultrasonography may be unavailable in resource-constrained settings, and delays may occur in emergencies when physicians are unfamiliar with its use [10,11].

A novel, easily identifiable external landmark would aid rapid and accurate chest tube insertion, particularly when conventional anatomical indicators are obscured or when ultrasound is unavailable. Although previous studies have suggested using the mid-arm point and the nipple level, these landmarks are unreliable, being unsuitable in patients with humeral fractures or highly variable among individuals, respectively [12,13]. Based on our preliminary research, we hypothesized that inserting a chest tube at the intersection of the midpoint of the full sternal length and the MAL may represent a safer and more reproducible alternative [14]. However, no study has directly compared these three alternative methods to determine the most optimal approach.

This study aimed to compare four anatomical landmark-based techniques for chest tube insertion: the conventional fifth ICS method, the mid-sternum method (based on the midpoint of the sternum), the nipple method, and the mid-arm point method.

## 2. Materials and Methods

### 2.1. Ethics Statement

This study was conducted following the principles of the Declaration of Helsinki and was approved by the Institutional Review Board (IRB approval No. GFIRB2025-096). The requirement for informed consent was waived due to the retrospective design of the study.

### 2.2. Study Design and Setting

This retrospective study was conducted at a single regional Level I trauma center affiliated with a tertiary university hospital in South Korea.

The study was based on preliminary data collected for submission to the Korea Trauma Data Bank (KTDB) at the same trauma center where the research was performed. The KTDB is a nationwide trauma registry established under the supervision of the Ministry of Health and Welfare of the Republic of Korea. All regional trauma centers are mandated to register clinical data for all patients with trauma, regardless of injury severity.

From the database, selected clinical variables—including age, sex, body weight, height, mechanism of injury, initial responsiveness, initial Glasgow Coma Scale (GCS) score, and final Injury Severity Score (ISS)—were extracted for analysis.

### 2.3. Inclusion and Exclusion Criteria

The study population comprised patients with trauma who visited the regional trauma center between February and June 2022 and met the following criteria: (1) adult patients aged 18 or older who were registered in the KTDB preliminary data during the study period, and (2) patients who underwent a “Chest Computed Tomography (CT) Trauma” scan.

Patients whose CT scans were performed with their arms elevated—potentially distorting anatomical structures—were excluded from the analysis.

### 2.4. CT Scanning Protocol

Patients with suspected chest trauma who presented to the trauma center underwent CT scans under the order “chest CT trauma”. Scans were performed with patients in the supine position and both arms positioned downward alongside the torso.

Non-contrast and intravenous contrast-enhanced CT scans of the chest, abdomen, and pelvis were acquired using a 128-channel scanner (Siemens Healthineers, Forchheim, Bavaria, Germany). Image acquisition commenced after a 6 s delay, with a total scan duration of 8.51 s. Scanning parameters included a pitch ratio of 1.5:1 and a gantry rotation time of 0.5 s. All images were retrospectively reconstructed using a CT console with a slice thickness of 3.00 mm.

### 2.5. Definitions

Details regarding term definitions and measurement processes are provided in the Appendix A.

MAL: This term was defined as in our previous study based on the work of Wax and Leibowitz [14,15]. On the axial CT image at the level of the xiphoid process, the peripheral point of the pleura was designated as the reference point (Figure 1A). A craniocaudal line drawn from this point was considered the MAL.Mid-sternum point: The midpoint of the total sternal length was calculated by summing the lengths of the manubrium and body–xiphoid segments of the sternum, which were measured separately (Figure 1B) [16].Mid-arm point: The midpoint of the linear distance between the acromion and olecranon processes was measured on the chest CT topogram (Figure 1C) [12].Nipple level: The term was defined as the level at which both nipples are visualized on the sagittal CT image (Figure 1D) [13].Fifth ICS: The term was defined as the space between the fifth and sixth ribs, identified on the sagittal image (Figure 1E)Methods and possible insertion sites used in this study: For each anatomical point described above, the corresponding level on the axial plane was identified. The point at which this level intersected the MAL was defined as the possible insertion site. Each method was named according to its anatomical reference point: mid-sternum, mid-arm point, nipple, and fifth ICS methods.Possible insertion route: The term was defined as a horizontal scout line passing through each possible insertion site.Eligibility for insertion within the safe triangle (safety zone): A possible insertion route was considered eligible if it did not intersect the pectoralis major and latissimus dorsi, and it is positioned superior to the fifth ICS level.

### 2.6. Outcomes and Measured Variables

The primary outcomes measured on the CT scans were as follows: (1) Variables were related to the possible insertion site: ICS or rib level at the insertion point; (2) Variables were related to the possible insertion route: perpendicular distance from the route to the pectoralis major muscle, whether the route penetrates the pectoralis major muscle, perpendicular distance from the route to the latissimus dorsi muscle, whether the route penetrates the latissimus dorsi muscle, chest wall thickness along the route, whether the route passes through breast tissue, distance from the route to the highest point of the diaphragm in mid-axillary coronal plane, cranio-caudal distance from the route to the highest point of the diaphragm in any coronal plane, whether the route penetrates the diaphragm, and whether the route is located within the anatomical boundaries of the safe triangle.

### 2.7. Statistical Analysis

Data were analyzed using SPSS software (version 22.0; SPSS Inc., Chicago, IL, USA). Categorical variables are presented as frequencies and percentages, and continuous variables are reported as medians and interquartile ranges (IQRs). Since comparisons were made within the same subjects, McNemar’s test was applied for categorical variables, and the Wilcoxon signed-rank test was used for continuous variables.

The conventional fifth ICS method was designated as the gold standard to evaluate the accuracy of each method. Based on this reference, sensitivity, specificity, positive predictive value (PPV), negative predictive value (NPV), and overall accuracy were calculated for each alternative insertion method. These measures and their confidence intervals were calculated using MedCalc^®^ software (version 20.106; MedCalc Software Ltd., Ostend, Belgium). All statistical analyses were two-sided, and a *p*-value < 0.05 was considered statistically significant.

## 3. Results

### 3.1. Characteristics of the Study Population

During the study period, 1016 trauma patients were registered in the KTDB (Figure 2). Of these, 771 were excluded because they did not meet the inclusion criteria, leaving 245 patients for analysis. The median age was 55.0 years (IQR, 42.0–64.0), and 186 patients (75.9%) were male (Table 1). The median body mass index (BMI) was 24.0 kg/m^2^ (IQR, 21.2–26.7).

Among the 245 patients, 38 (15.5%) had sternal fractures, 70 (28.6%) had right-sided rib fractures, 68 (27.8%) had left-sided rib fractures, five (2.0%) had right humeral fractures, and two (0.8%) had left humeral fractures. Blunt trauma was the mechanism of injury in 227 patients (92.7%). The median revised trauma score was 12.0 (IQR, 11.0–12.0), and the median ISS was 17.0 (IQR, 10.0–26.0).

The total length of the sternum was 199.2 mm (IQR, 183.1–212.9), and the length of its lower half was 99.6 mm (IQR, 91.6–106.4).

### 3.2. Comparison Between the Landmark-Based Techniques for Tube Thoracostomy (TT)

Table 2 presents a comparison of landmark-based methods for the right thorax. The fifth ICS was identified as a possible insertion site in 124 (50.6%), 102 (41.6%), and 99 (40.4%) patients using the mid-sternum, nipple, and mid-arm point methods, respectively. The number of patients whose insertion routes passed through the fifth ICS or higher was 236 (96.3%) for the mid-sternum method, 115 (46.9%) for the nipple method, and 216 (88.2%) for the mid-arm point method.

The TT route did not pass through the diaphragm in 201 patients (82.0%) using the fifth ICS method, in 227 (92.7%) using the mid-sternum method, in 136 (55.5%) using the nipple method, and in 221 (90.2%) using the mid-arm point method. Statistically significant differences were observed for comparisons with the fifth ICS method (*p* < 0.001).

The TT route was located within the safety zone in 214 (87.3%) patients using the fifth ICS method and in 202 (82.4%), 102 (41.6%), and 186 (75.9%) patients using the mid-sternum, nipple, and mid-arm point methods, respectively. Statistically significant differences were observed when compared with the fifth ICS method (*p* = 0.031, *p* < 0.001, and *p* = 0.003, respectively).

The number of patients whose routes were located within the safety zone and did not pass through the diaphragm was 174 (71.0%) using the fifth ICS method, 185 (75.5%) using the mid-sternum method, 60 (24.5%) using the nipple method, and 175 (71.4%) using the mid-arm point method. Compared to the fifth ICS method, there were no statistically significant differences for the mid-sternum (*p* = 0.185) and mid-arm point (*p* = 0.596) methods, whereas a significant difference was observed for the nipple method (*p* < 0.001).

Table 3 presents a comparison of landmark-based methods for the left thorax. The fifth ICS was identified as a possible insertion site in 131 (53.5%), 110 (44.9%), and 81 (33.1%) patients using the mid-sternum, nipple, and mid-arm point methods, respectively. The number of patients whose routes passed through the fifth ICS or higher was 228 (93.1%) for the mid-sternum method, 131 (53.5%) for the nipple method, and 223 (91.0%) for the mid-arm point method.

The TT route did not pass through the diaphragm in 239 patients (97.6%) using the fifth ICS method, 242 (98.8%) using the mid-sternum method, 213 (86.9%) using the nipple method, and 233 (95.1%) using the mid-arm point method. Compared to the fifth ICS method, there were no statistically significant differences for the mid-sternum (*p* = 0.375) and mid-arm point (*p* = 0.125) methods, whereas a significant difference was observed for the nipple method (*p* < 0.001).

The TT route was located within the safety zone in 218 patients (89.0%) using the fifth ICS method, 201 patients (82.0%) using the mid-sternum method, 123 patients (50.2%) using the nipple method, and 193 patients (78.8%) using the mid-arm point method; statistically significant differences were observed for all comparisons with the fifth ICS method (*p* = 0.001, *p* < 0.001, and *p* = 0.018, respectively).

The number of patients whose routes were within the safety zone and did not pass through the diaphragm was 212 (86.5%) for the fifth ICS method, 199 (81.2%) for the mid-sternum method, 114 (46.5%) for the nipple method, and 193 (78.8%) for the mid-arm point method. Compared with the fifth ICS method, there was no statistically significant difference for the mid-arm point method (*p* = 0.243), whereas significant differences were observed for the mid-sternum (*p* = 0.026) and nipple (*p* < 0.001) methods.

### 3.3. Safety Performance of the Landmark-Based Techniques for Tube Thoracostomy

Table 4 and Table 5 summarize the safety performance of each landmark-based technique evaluated using the fifth ICS method as a reference.

In the right thorax, the sensitivities for routes located within the safety zone were 91.1%, 45.3%, and 85.0% for the mid-sternum, nipple, and mid-arm point methods, respectively; the corresponding accuracies were 89.4%, 50.2%, and 82.3%, respectively. The sensitivities for routes that did not pass through the diaphragm were 96.5% for both the mid-sternum and mid-arm point methods, and 63.7% for the nipple method. For routes located within the safety zone and not passing through the diaphragm, the sensitivities were 86.8% using the mid-sternum method, 28.2% using the nipple method, and 84.7% using the mid-arm point method; the corresponding accuracies were 76.7%, 44.5%, and 76.0%, respectively.

In the left thorax, the sensitivities for routes within the safety zone were 90.4%, 53.2%, and 89.4% for the mid-sternum, nipple, and mid-arm point methods, respectively; the corresponding accuracies were 89.8%, 55.5%, and 87.2%, respectively.

The sensitivities for routes not passing through the diaphragm were 99.6% for both the mid-sternum and mid-arm point methods and 88.7% for the nipple method; the respective accuracies were 98.0%, 88.6%, and 97.0%, respectively.

For routes located within the safety zone and not passing through the diaphragm, the sensitivities were 90.1% for the mid-sternum method, 50.0% for the nipple method, and 89.1% using the mid-arm point method; the corresponding accuracies are 88.2%, 53.5%, and 84.6%, respectively.

## 4. Discussion

In this study comparing landmark-based techniques for TT placement—using the fifth ICS method as the reference standard—the mid-sternum method was found to be the most reliable, followed by the mid-arm point and nipple methods. However, the accuracy of the mid-sternum method in locating routes within the safety zone was lower on both sides of the thorax than that of the fifth ICS method. Most performance metrics of the mid-sternum method, with regard to routes located within the safety zone, were satisfactory. Compared with the fifth ICS method, the mid-sternum approach offered greater safety against diaphragmatic injury on the right side of the thorax, while no significant difference was observed on the left.

Although the fifth ICS method is widely accepted as the conventional approach for TT placement, it has limitations [2]. First, for novice practitioners, accurately identifying the safety zone can be challenging and may lead to procedural complications. Griffiths et al. reported that 45% of junior doctors could not accurately locate the safety zone [17]. In another study involving patients with trauma, 17 complications were reported in 76 TT insertions performed by residents [18]. These findings highlight that, even with the fifth ICS method, determining the correct insertion site is not always straightforward. Moreover, in emergency situations, clinicians may find it even more difficult to identify anatomical landmarks within the safety zone, potentially increasing procedure time or the risk of inserting the tube below the diaphragm.

The diaphragm is attached to the xiphoid process of the sternum, ribs, and the lumbar vertebrae [19]. As the sternum lies superior to the diaphragm and has a fixed anatomical structure, it serves as a reliable landmark to avoid diaphragmatic injury. In addition, the sternum is the most prominent anterior thoracic structure, allowing clinicians to locate it quickly and intuitively. Unlike the fifth ICS—whose position can vary depending on the patient’s arm position or respiratory cycle—the length of the sternum remains constant and is unaffected by patient positioning. For these reasons, we consider the mid-sternum method a safe and straightforward alternative to the conventional fifth ICS method. In our study, when comparing existing landmark-based techniques using the fifth ICS method as the reference, the mid-sternum method demonstrated a sensitivity of up to 99.6% for some variables, outperforming any other alternative technique developed to date. Given its anatomical stability and high accuracy, the mid-sternum method appears to be user-friendly, even for less experienced practitioners.

Second, even when using the fifth ICS method, various organ injuries have been reported [20]. Injuries to structures superior to the diaphragm—such as the lungs, esophagus, and heart—can occur but may be avoided through careful control of the insertion depth and direction and precise identification of the MAL. However, diaphragmatic or intra-abdominal organ injuries may still occur even when anatomical landmarks are properly identified. According to Kwiatt et al., intra-abdominal placement of a TT occurs in fewer than 1% of cases; however, approximately one-third of these result in bowel injuries [20]. In our study, routes based on the fifth ICS method crossed the diaphragm in 18% of right-sided insertions and 3% of left-sided insertions. Thus, the fifth ICS method is not a perfect safeguard against diaphragmatic or subdiaphragmatic injuries. In contrast, fewer routes crossed the diaphragm when the mid-sternum method was used, suggesting that it may reduce the risk of such injuries.

Therefore, the mid-sternum method may serve as a useful alternative for performing TT quickly and safely in emergency settings, helping to avoid diaphragmatic injury. However, this method has some limitations. First, its use may be restricted in cases of sternal fractures. In our study population, 15.5% of patients had sternal fractures. Whether this constitutes a limitation or highlights the robustness of our findings remains unclear. Nonetheless, even among patients with a potentially shortened sternal length due to fracture, the mid-sternum method resulted in fewer routes crossing the diaphragm than the fifth ICS method. Further studies focusing exclusively on patients without sternal fractures are warranted. Second, the number of routes located within the safety zone was slightly lower in the mid-sternum method than in the fifth ICS method. However, even with the fifth ICS method, 13% of the right-sided and 11% of left-sided routes fell outside the safety zone. This finding suggests that the fifth ICS method also carries a risk of muscle injury, such as the pectoralis major and latissimus dorsi. Therefore, both methods present a risk of muscle injury in some patients. While minimizing injury is always ideal, minor damage to surrounding muscles may be clinically acceptable in emergency scenarios.

As discussed above, the mid-sternum method is not superior in all clinical situations. The following guide may help in selecting an appropriate insertion method. In general, in clinical practice, clinicians should prioritize the conventional 5th ICS method. If possible, they should use ultrasound guidance because the conventional method has limitations. In emergency situations, clinicians may choose the following approaches: first, clinicians may use the mid-sternum method as an alternative for patients with distorted thoracic anatomy, such as severe subcutaneous emphysema or multiple rib fractures. Second, in obese patients, clinicians may prefer the mid-sternum method because palpating the ribs is difficult due to thick subcutaneous fat. Third, in female patients, nipple position can be inconsistent, so clinicians may rely on the sternum as a more reliable landmark than the nipple. Fourth, in patients with sternal fractures, the sternum length may be altered. Clinicians should use caution when applying the mid-sternum method. The mid-arm point method may serve as an ancillary landmark, but clinicians should exercise caution if a humeral fracture or shoulder dislocation is suspected. Finally, the routes using the nipple method were often located at or below the 6th ICS in our study. Clinicians should avoid using it as a standalone landmark. Ultimately, further studies are needed to establish formal guidelines.

### Limitations

This study has some limitations. First, it was conducted at a single center with a relatively small sample size. Second, there may be differences between measurements obtained from actual patients and those derived from CT images. Although the sternum is not perfectly straight, we measured it as a linear structure, under the assumption that clinicians perceive it as such during TT procedures. Further prospective studies are required to address this limitation. Third, during TT insertion, clinicians often direct the tube cephalad. However, in this study, we measured the distance to the diaphragm as a transverse route from the insertion site, which may have led to an overestimation of diaphragmatic contact compared to real-world settings. Fourth, in patients with altered mental status, irregular respiration patterns may have affected the accuracy of our CT-based assessments.

## 5. Conclusions

In this study comparing landmark-based techniques for TT placement—using the fifth ICS method as the reference standard—the mid-sternum method was found to be the most reliable, followed by the mid-arm point and nipple methods. The mid-sternum method appears to be a safe, feasible, and easy-to-use alternative to the fifth ICS method for TT insertion. It demonstrated superior safety with respect to diaphragmatic injury, particularly on the right side. Given its anatomical consistency and overall performance, the mid-sternum method may serve as a practical reference point for clinicians, particularly in emergency settings or when conventional anatomical landmarks are obscured.

## Figures and Tables

**Figure 1 jcm-14-07571-f001:**
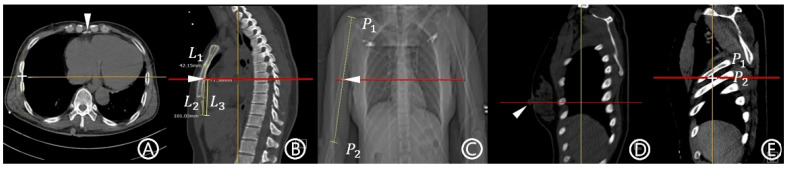
Anatomical reference points and measurement methods were used to define potential chest tube insertion levels based on computed tomography (CT) images. (**A**) An axial CT image illustrating the method for defining the mid-axillary line (MAL). The most peripheral point of the pleura at the level of the xiphoid process (arrowhead) is marked with a cross (+). A cranio-caudal scout line was drawn vertically from this point and designated as the MAL. The horizontal yellow line corresponds to the automatically generated scout line from the cross (+) point. (**B**) A sagittal CT image demonstrating how the mid-sternum level was determined. The length of the manubrium (L1) and the body–xiphoid segment (L2) were measured, and their sum defined the total sternal length. The midpoint of this length was identified as the mid-sternum point (arrowhead). L3 denotes the length of the lower half of the sternum. The mid-sternum level is indicated by a red horizontal line, and the yellow vertical line represents the MAL. (**C**) A chest CT topogram showing the method for determining the mid-arm level. The humeral length was measured from the acromion process (P1) to the olecranon (P2), and the midpoint (arrowhead) was designated as the mid-arm point. The mid-arm level is indicated by a red horizontal line. (**D**) A sagittal CT image illustrating the nipple level. The right nipple, marked with an arrowhead, served as the anatomical reference point. The nipple level is indicated by a red horizontal line, and the yellow vertical line represents the MAL. (**E**) A sagittal CT image illustrating the anatomical definition of the fifth intercostal space (ICS) level. The fifth rib (P1) and sixth rib (P2) are identified, and the space between them is marked with a cross (+) to indicate the fifth ICS. The red horizontal line represents the fifth ICS level, and the yellow vertical line indicates the MAL.

**Figure 2 jcm-14-07571-f002:**
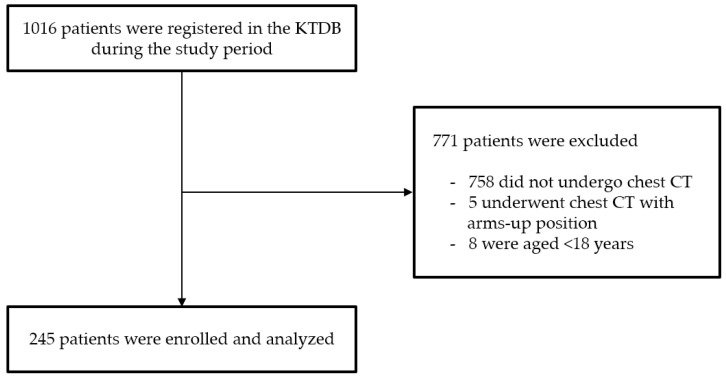
Study flow chart. Abbreviations: KTDB, Korea Trauma Data Bank; CT, computed tomography.

**Table 1 jcm-14-07571-t001:** Baseline characteristics of the enrolled patients (n = 245).

Variable	
Age (years)	55.0 (42.0–64.0)
Sex (male)	186 (75.9)
ASA PS classification	
I	100 (40.8)
II	70 (28.6)
III	27 (11.0)
IV	47 (19.2)
V	1 (0.4)
VI	0 (0.0)
Body mass index (kg/m^2^)	24.0 (21.2–26.7)
Weight (kg)	67.4 (56.9–78.0)
Height (cm)	168.0 (160.0–174.0)
Sternum fracture	38 (15.5)
Rib fracture	
Right	70 (28.6)
Left	68 (27.8)
Multiple rib fracture	
Right	55 (22.4)
Left	60 (24.5)
Number of fractured ribs	
Right	5.0 (2.0–6.0)
Left	4.0 (2.0–6.0)
Humerus fracture	
Right	5 (2.0)
Left	2 (0.8)
Subcutaneous emphysema	
Right	22 (9.0)
Left	22 (9.0)
Pneumothorax	
Right	25 (10.2)
Left	21 (8.6)
Hemothorax	
Right	27 (11.0)
Left	25 (10.2)
Injury mechanism (blunt injury)	227 (92.7)
Consciousness at admission to the ED	
Alert	132 (53.9)
Verbal response	49 (20.0)
Painful response	37 (15.1)
Unresponsive	27 (11.0)
Vital signs at admission to the ED	
Systolic blood pressure (mmHg)	141.0 (120.0–163.0)
Pulse rate (beats/min)	88.0 (75.0–106.0)
Respiration rate (breaths/min)	20.0 (19.0–24.0)
Body temperature (℃)	36.5 (36.0–36.9)
Peripheral capillary O_2_ saturation (%)	97.0 (95.0–99.0)
Total GCS score	14.0 (11.0–15.0)
Revised trauma score	12.0 (11.0–12.0)
Final ISS	17.0 (10.0–26.0)
Total length of the sternum (mm)	199.2 (183.1–212.9)
Length of the manubrium	50.6 (46.9–53.9)
Length of the sternum body–xiphoid process	147.9 (134.4–159.1)
Length of the lower half of the sternum (mm)	99.6 (91.6–106.4)

Values are presented as median (interquartile range) or number (%). ASA PS, American Society of Anesthesiologists physical status; ED, emergency department; GCS, Glasgow Coma Scale; ISS, Injury Severity Score.

**Table 2 jcm-14-07571-t002:** Analysis of anatomical structures and pathways during thoracostomy using landmark techniques (right).

	Fifth ICS Method	Mid-Sternum Method	*p*-Value ^a^	Nipple Method	*p*-Value ^b^	Mid-Arm Point Method	*p*-Value ^c^
ICS			N/C		N/C		N/C
2nd						1 (0.4)	
3rd		7 (2.9)				14 (5.7)	
4th		105 (42.9)		13 (5.3)		102 (41.6)	
5th	245 (100.0)	124 (50.6)		102 (41.6)		99 (40.4)	
6th		9 (3.7)		123 (50.2)		19 (7.8)	
7th				7 (2.9)		2 (0.8)	
8th							
Level of the fifth ICS and above		236 (96.3)	N/C	115 (46.9)	N/C	216 (88.2)	N/C
CWT (mm)	40.1 (32.7–50.1)	40.8 (32.7–49.6)	0.855	38.4 (31.4–48.9)	0.577	40.2 (32.9–48.1)	0.809
Distance to the pectoralis major (mm)	38.7 (32.1–46.3)	38.0 (30.4–44.3)	0.195	46.3 (37.0–53.3)	<0.001	38.3 (29.4–45.7)	0.039
Not passing the pectoralis major	245 (100.0)	245 (100.0)	N/C	245 (100.0)	N/C	237 (96.7)	N/C
Distance to the latissimus dorsi (mm)	20.6 (11.7–32.3)	18.0 (9.1–30.3)	<0.001	21.6 (12.5–31.4)	0.002	18.3 (7.7–29.9)	<0.001
Not passing the latissimus dorsi	214 (87.3)	206 (84.1)	0.134	215 (87.8)	1.000	205 (83.7)	1.000
Distance to the highest diaphragm in the MAL (mm)	30.9 (8.3–54.1)	36.7 (21.9–55.5)	<0.001	8.4 (0.0–29.3)	<0.001	39.7 (17.4–60.6)	<0.001
CC distance to the highest diaphragm in any AL (mm)	28.6 (7.6–49.7)	33.5 (18.3–52.6)	<0.001	6.1 (0.0–25.3)	<0.001	34.7 (15.2–58.4)	<0.001
Not passing the diaphragm	201 (82.0)	227 (92.7)	<0.001	136 (55.5)	<0.001	221 (90.2)	<0.001
Not passing breast tissue	241 (98.4)	243 (99.2)	0.500	243 (99.2)	0.687	233 (95.1)	1.000
In the safety zone	214 (87.3)	202 (82.4)	0.031	102 (41.6)	<0.001	186 (75.9)	0.003
In the safety zone and not passing the diaphragm	174 (71.0)	185 (75.5)	0.185	60 (24.5)	<0.001	175 (71.4)	0.596
In the safety zone and not passing either breast tissue or the diaphragm	174 (71.0)	183 (74.7)	0.281	60 (24.5)	<0.001	173 (70.6)	0.787

Values are presented as median (interquartile range) or number (%). ICS, intercostal space; N/C, not calculated; CWT, chest wall thickness; MAL, mid-axillary line; CC, cranio-caudal; AL, axillary line. ^a^ Comparison between the mid-sternum method and the standard fifth ICS method. ^b^ Comparison between the nipple method and the standard fifth ICS method. ^c^ Comparison between the mid-arm point method and the standard fifth ICS method.

**Table 3 jcm-14-07571-t003:** Analysis of anatomical structures and pathways during thoracostomy using landmark techniques (left).

	Fifth ICS Method	Mid-Sternum Method	*p*-Value ^a^	Nipple Method	*p*-Value ^b^	Mid-Arm Point Method	*p*-Value ^c^
ICS			N/C		N/C		N/C
2nd							
3rd		9 (3.7)				29 (11.8)	
4th		88 (35.9)		21 (8.6)		113 (46.1)	
5th	245 (100.0)	131 (53.5)		110 (44.9)		81 (33.1)	
6th		16 (6.5)		106 (43.3)		10 (4.1)	
7th		1 (0.4)		8 (3.3)		1 (0.4)	
8th							
Level of the fifth ICS and above		228 (93.1)	N/C	131 (53.5)	N/C	223 (91.0)	N/C
CWT (mm)	39.0 (31.4–47.6)	39.9 (31.6–48.2)	0.001	39.3 (29.9–47.4)	0.771	41.1 (31.9–47.7)	<0.001
Distance to the pectoralis major (mm)	35.7 (29.0–44.2)	33.6 (28.2–41.3)	0.012	43.5 (35.7–52.6)	<0.001	35.1 (27.1–41.8)	0.005
Not passing the pectoralis major	245 (100.0)	245 (100.0)	N/C	245 (100.0)	N/C	234 (95.5)	N/C
Distance to the latissimus dorsi (mm)	24.8 (13.9–33.9)	21.7 (13.0–32.7)	0.031	24.5 (14.6–34.7)	<0.001	19.0 (10.7–30.7)	<0.001
Not passing the latissimus dorsi	218 (89.0)	216 (88.2)	0.774	228 (93.1)	0.041	203 (82.9)	0.503
Distance to the highest diaphragm in the MAL (mm)	49.2 (30.6–69.9)	54.0 (38.2–72.4)	<0.001	29.6 (12.9–46.2)	<0.001	67.3 (47.7–84.3)	<0.001
CC distance to the highest diaphragm in any AL (mm)	49.2 (30.3–68.3)	52.1 (37.7–70.4)	<0.001	27.4 (11.6–45.2)	<0.001	64.4 (46.6–80.3)	<0.001
Not passing the diaphragm	239 (97.6)	242 (98.8)	0.375	213 (86.9)	<0.001	233 (95.1)	0.125
Not passing breast tissue	241 (98.4)	240 (98.0)	1.000	241 (98.4)	1.000	231 (94.3)	1.000
In the safety zone	218 (89.0)	201 (82.0)	0.001	123 (50.2)	<0.001	193 (78.8)	0.018
In the safety zone and not passing the diaphragm	212 (86.5)	199 (81.2)	0.026	114 (46.5)	<0.001	193 (78.8)	0.243
In the safety zone and not passing either breast tissue or the diaphragm	208 (84.9)	194 (79.2)	0.026	114 (46.5)	<0.001	192 (78.4)	0.511

Values are presented as median (interquartile range) or number (%). ICS, intercostal space; N/C, not calculated; CWT, chest wall thickness; MAL, mid-axillary line; CC, cranio-caudal; AL, axillary line. ^a^ Comparison between the mid-sternum method and the standard fifth ICS method. ^b^ Comparison between the nipple method and the standard fifth ICS method. ^c^ Comparison between the mid-arm point method and the standard fifth ICS method.

**Table 4 jcm-14-07571-t004:** Prediction of the safety performance of the mid-sternum, nipple, and mid-arm point methods based on the fifth intercostal space method (right).

		Fifth ICS Method		Sensitivity (%)	Specificity (%)	PPV (%)	NPV (%)	Accuracy %)	*p*-Value
		Safe	Unsafe	CI	CI	CI	CI	CI	
Locating in the safety zone									
Mid-sternum method	Safe	195	7	91.1	77.4	96.5	55.8	89.4	0.031
	Unsafe	19	24	86.5–94.6	58.9–90.4	93.5–98.2	44.1–66.9	84.8–93.0	
Nipple method	Safe	97	5	45.3	83.9	95.1	18.2	50.2	<0.001
	Unsafe	117	26	38.5–52.3	66.3–94.5	89.6–97.8	15.4–21.3	43.8–56.6	
Mid-arm point method	Safe	175	11	85.0	64.5	94.1	39.2	82.3	0.003
	Unsafe	31	20	79.3–89.5	45.4–80.8	90.8–96.2	29.8–49.5	76.8–86.9	
Not passing the diaphragm									
Mid-sternum method	Safe	194	33	96.5	25.0	85.5	61.1	83.7	<0.001
	Unsafe	7	11	93.0–98.6	13.2–40.3	83.2–87.5	39.2–79.3	78.4–88.1	
Nipple method	Safe	128	8	63.7	81.8	94.1	33.0	66.9	<0.001
	Unsafe	73	36	56.6–70.3	67.3–91.8	89.4–96.8	28.2–38.3	60.7–72.8	
Mid-arm point method	Safe	190	31	96.5	22.5	86.0	56.3	84.0	<0.001
	Unsafe	7	9	92.8–98.6	10.8–38.5	83.8–87.9	33.7–76.5	78.7–88.4	
Locating in the safety zone and not passing the diaphragm									
Mid-sternum method	Safe	151	34	86.8	52.1	81.6	61.7	76.7	0.185
	Unsafe	23	37	80.8–91.4	39.9–64.1	77.6–85.1	50.9–71.4	70.9–81.9	
Nipple method	Safe	49	11	28.2	84.5	81.7	32.4	44.5	<0.001
	Unsafe	125	60	21.6–35.5	74.0–92.0	71.1–89.0	29.5–35.5	38.2–51.0	
Mid-arm point method	Safe	144	31	84.7	53.7	82.3	58.1	76.0	0.596
	Unsafe	26	36	78.4–89.8	41.1–66.0	78.1–85.8	47.7–67.8	70.0–81.2	
Locating in the safety zone, not passing the diaphragm, and not passing breast tissue									
Mid-sternum method	Safe	151	32	86.8	54.9	82.5	62.9	77.6	0.281
	Unsafe	23	39	80.8–91.4	42.7–66.8	78.4–86.0	52.3–72.4	71.8–82.6	
Nipple method	Safe	49	11	28.2	84.5	81.7	32.4	44.5	<0.001
	Unsafe	125	60	21.6–35.5	74.0–92.0	71.1–89.0	29.5–35.5	38.2–51.0	
Mid-arm point method	Safe	144	29	84.7	56.7	83.2	59.4	76.8	0.787
	Unsafe	26	38	78.4–89.8	44.0–68.8	78.9–86.8	49.2–68.8	70.9–82.0	

Values are presented as numbers. ICS, intercostal space; PPV, positive predictive value; NPV, negative predictive value; CI, confidence interval.

**Table 5 jcm-14-07571-t005:** Prediction of the safety performance of the mid-sternum, nipple, and mid-arm point methods based on the fifth intercostal space method (left).

		Fifth ICS Method		Sensitivity (%)	Specificity (%)	PPV (%)	NPV (%)	Accuracy (%)	*p*-Value
		Safe	Unsafe	CI	CI	CI	CI	CI	
Locating in the safety zone									
Mid-sternum method	Safe	197	4	90.4	85.2	98.0	52.3	89.8	0.001
	Unsafe	21	23	85.7–93.9	66.3–95.8	95.2–99.2	41.5–62.9	85.3–93.3	
Nipple method	Safe	116	7	53.2	74.1	94.3	16.4	55.5	<0.001
	Unsafe	102	20	46.4–60.0	53.7–88.9	89.6–96.9	13.1–20.3	49.0–61.8	
Mid-arm point method	Safe	185	8	89.4	70.4	95.9	46.3	87.2	0.018
	Unsafe	22	19	84.4–93.2	49.8–86.2	92.8–97.6	35.2–57.9	82.2–91.2	
Not passing the diaphragm									
Mid-sternum method	Safe	238	4	99.6	33.3	98.4	66.7	98.0	0.375
	Unsafe	1	2	97.7–100.0	4.3–77.7	97.1–99.1	17.3–95.0	95.3–99.3	
Nipple method	Safe	212	1	88.7	83.3	99.5	15.6	88.6	<0.001
	Unsafe	27	5	84.0–92.4	35.9–99.6	97.3–99.9	10.1–23.5	83.9–92.3	
Mid-arm point method	Safe	227	6	99.6	0.0	97.4	0.0	97.0	0.125
	Unsafe	1	0	97.6–100.0	0.0–45.9	97.4–97.4	N/C	93.9–98.8	
Locating in the safety zone and not passing the diaphragm									
Mid-sternum method	Safe	191	8	90.1	75.8	96.0	54.4	88.2	0.026
	Unsafe	21	25	85.3–93.8	57.7–88.9	92.9–97.8	43.2–65.1	83.4–91.9	
Nipple method	Safe	106	8	50.0	75.8	93.0	19.1	53.5	<0.001
	Unsafe	106	25	43.1–56.9	57.7–88.9	87.7–96.1	15.7–23.0	47.0–59.8	
Mid-arm point method	Safe	179	14	89.1	57.6	92.8	46.3	84.6	0.243
	Unsafe	22	19	83.9–93.0	39.2–74.5	89.5–95.0	34.6–58.5	79.3–89.0	
Locating in the safety zone, not passing the diaphragm, and not passing breast tissue									
Mid-sternum method	Safe	184	10	88.5	73.0	94.9	52.9	86.1	0.026
	Unsafe	24	27	83.3–92.5	55.9–86.2	91.5–96.9	42.4–63.2	81.2–90.2	
Nipple method	Safe	106	8	51.0	78.4	93.0	22.1	55.1	<0.001
	Unsafe	102	29	44.0–57.9	61.8–90.2	87.6–96.1	18.6–26.1	48.6–61.4	
Mid-arm point method	Safe	176	16	89.3	56.8	91.7	50.0	84.2	0.511
	Unsafe	21	21	84.2–93.3	39.5–72.9	88.3–94.1	37.9–62.1	78.9–88.6	

Values are presented as numbers. ICS, intercostal space; PPV, positive predictive value; NPV, negative predictive value; CI, confidence interval; N/C, not calculated.

## Data Availability

The datasets used and/or analyzed during the current study are available from the corresponding author upon reasonable request.

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
