# Peer review of "Determination of the Optimal Landmark for Tube Thoracostomy in Trauma Patients: A Retrospective Study"

_jcm, 2025, doi:10.3390/jcm14217571_

Round 1
Reviewer 1 Report
Comments and Suggestions for Authors
I was pleased to review the authors' work entitled "Determination of the Optimal Landmark for Tube Thoracostomy in Trauma Patients: A Comparison of Landmark Techniques Using Chest Computed Tomography, Including a Novel Mid-Sternum Method". This retrospective study investigates the accuracy and safety of landmark-based thoracostomy (TT) techniques, including a novel mid-sternum method. The authors concluded that the mid-sternum method demonstrated high anatomical safety and performance comparable to or superior to the conventional fifth intercostal space (ICS) method, particularly in minimizing the risk of diaphragmatic injury. The manuscript is well-written and the topic is interesting. In my opinion, it pending some minor corrections:
1) The title "Determination of the Optimal Landmark for Tube Thoracostomy in Trauma Patients: A Comparison of Landmark Techniques Using Chest Computed Tomography, Including a Novel Mid-Sternum Method" is very extensive. It could be just "Determination of the Optimal Landmark for Tube Thoracostomy in Trauma Patients: A Retrospective Study".
2) Lines 62-64: Reference is missing
3) In the introduction section, I would suggest using the current literature based on recently published articles. Reference 1 (2011), Reference 3 (2010). Please use articles published in the last ten years
4) In the results section of the abstract, please add some information about the study population, such as age and gender.
5) In Table 1, could you please add information about the American Society of Anesthesiologists Physical Status (ASA-PS) score?
6) The discussion is very brief. Discuss the role of fresh frozen plasma pleurodesis as an effective treatment for pneumothorax.
Reviewer 2 Report
Comments and Suggestions for Authors
This study systematically compares four thoracostomy tube insertion approaches through computed tomography image analysis. It innovatively proposes the sternal midline approach as a safer alternative to the conventional fifth intercostal space method and provides substantial imaging evidence to support its safety and reliability. The research design is scientific, the data analysis is comprehensive, and the findings hold significant clinical guidance value, particularly in emergency settings and resource-limited environments. The authors objectively analyze both the advantages and limitations of the method, demonstrating a scientific assessment attitude. The manuscript is of high quality. Specific suggestions for revision are as follows:
- Although the number of safe needle paths in the sternal midline approach is slightly lower than that of the fifth intercostal space method, the clinical implications of this difference should be further discussed. It is recommended to provide comparative data on the distribution of needle paths near vital organs for both methods.
- Confidence intervals for sensitivity and specificity calculations should be provided to enhance the reliability of the results. Additionally, the specific statistical tests used should be clearly stated.
- A brief description of key measurement methods (e.g., definition of the mid-axillary line, determination of the sternal midpoint) should be included in the main text, even if detailed methodologies are provided in the supplementary materials.
- Specific operational recommendations could be provided for special populations (e.g., obese patients, female patients, patients with sternal fractures).
- Based on the findings, a clear clinical decision-making process should be proposed, indicating when to prioritize the sternal midline approach.
